# Fabrication, Optimization, and Characterization of Antibacterial Electrospun Shellac Fibers Loaded with *Kaempferia parviflora* Extract

**DOI:** 10.3390/pharmaceutics15010123

**Published:** 2022-12-29

**Authors:** Wantanwa Krongrawa, Sontaya Limmatvapirat, Mont Kumpugdee Vollrath, Prasat Kittakoop, Supachai Saibua, Chutima Limmatvapirat

**Affiliations:** 1Department of Industrial Pharmacy, Faculty of Pharmacy, Silpakorn University, Nakhon Pathom 73000, Thailand; 2Pharmaceutical Biopolymer Group (PBiG), Faculty of Pharmacy, Silpakorn University, Nakhon Pathom 73000, Thailand; 3Laboratory Pharmaceutical Technology, Faculty II-Mathematics-Physics-Chemistry, Berliner Hochschule für Technik, Luxemburger Str.10, 13353 Berlin, Germany; 4Chulabhorn Graduate Institute, Program in Chemical Sciences, Chulabhorn Royal Academy, Bangkok 10210, Thailand; 5Chulabhorn Research Institute, Kamphaeng Phet 6 Road, Laksi, Bangkok 10210, Thailand; 6Center of Excellence on Environmental Health and Toxicology (EHT), OPS, Ministry of Higher Education, Science, Research and Innovation, Bangkok 10400, Thailand; 7Bangkok Lab & Cosmetic Co., Ltd., 48/1, Moo 5, Nongsaesao Road, Tambon Namphu, Amphoe Muang Ratchaburi, Ratchaburi 70000, Thailand

**Keywords:** electrospun fiber, shellac, *Kaempferia parviflora*, Box–Behnken design, dissolution, methoxyflavones, antibacterial activity

## Abstract

This study aimed to develop a *Kaempferia parviflora* (KP) extract based on electrospun shellac fibers capable of transporting methoxyflavones. This study used a Box–Behnken design to determine the optimal production parameters that influence the fiber diameter and bead-to-fiber ratio responses. The optimization step produced fibers with a small diameter (574 nm) and a lower bead-to-fiber ratio (0.48 beads per fiber) by combining 37.25% *w*/*w* shellac and 1.50% *w*/*w* KP extract with a solution feed rate of 0.8 mL/h and an electrical voltage of 18 kV. The KP extract was found to be dispersed throughout the electrospun shellac fibers during the characterization study. The results were highly correlated with the theoretical values, indicating that the regression models used to predict the response variables were adequate. A study of in vitro dissolution confirmed that KP extract-loaded electrospun shellac fibers could produce a sustained-release profile within 10 h. Additionally, KP-infused shellac fibers demonstrated antibacterial activity against *Staphylococcus aureus*. This KP loading method combined with shellac properties provided a new delivery system and could be used to explore novel biomedical materials.

## 1. Introduction

*Kaempferia parviflora* (KP), also known as Kra-Chai-Dam, is frequently used in traditional medicine to treat inflammation, swelling, wounds, bacterial infection, ringworm, and a variety of other disorders [1]. Numerous findings regarding the biological activities of its rhizomes have been reported in literature reviews, including antimicrobial, antifungal, antioxidant, and anti-inflammatory activities [1,2,3,4,5]. The KP rhizome extract contained bioactive methoxyflavones, primarily 5,7-dimethoxyflavone (DMF), 5,7,4′-trimethoxyflavone (TMF), and 3,5,7,3′,4′-pentamethoxyflavone (PMF) (Figure 1), all of which exhibited favorable physicochemical properties [5,6], including a molecular weight of less than 500 Da, a melting point of between 150 and 200 °C, and a logarithm of octanol–water partition coefficient (log *P*) of approximately 2–3. Consequently, it might be an appropriate herbal active ingredient for the development of transdermal delivery products, such as solid lipid nanoparticles loaded with KP dichloromethane extract [6], monolithic drug-in-adhesive type patches containing KP ethanolic extract [7], and isopropyl myristate-based vehicles containing KP ethanolic extract [8].

Electrospun fibers with a high surface area-to-volume ratio and an excellent pore interconnection were well suited for a variety of advanced applications, particularly drug delivery [9]. Electrospinning was a widely applicable technique for increasing the permeability of bioactive compounds through the skin, and a variety of low-solubility active compounds could be loaded into the fibers for controlled release [10,11,12]. Electrospun fibers have a large surface area that comes into contact with biological fluids, resulting in a faster dissolution rate of active chemicals. However, the release of active compounds might well be altered regardless of the type of polymer used to fabricate electrospun fibers. Due to the mechanical properties, controlled degradation rate, and biological compatibility of shellac, a natural polymer derived from the excretions of the lac *Kerria lacca* insect, electrospun fibers seemed to become widely attractive [13,14,15,16,17]. Shellac resin contains derivatives of jalaric and laccijalaric acids (Figure 2), and a detailed analysis of the compounds in shellac was recently revealed by flow injection and liquid chromatography coupled with electrospray ionization and mass spectrometry [18].

The conductivity, surface tension, and viscosity of the electrospun solution; applied voltage, feed rate, and distance between the tip and collector of the electrospinning system; and humidity and temperature of the surrounding environment all affected the appearance of electrospun fibers [19]. In our previous study [20], we found that the shellac concentration was the most important parameter influencing the fiber diameter, while the applied voltage and solution flow rate were minor factors. Additionally, the shellac concentration and applied voltage had an effect on the formation of the beaded electrospun fibers. The use of shellac in the production of electrospun fibers has rarely been studied. One of the objectives of this study was to ascertain the feasibility of electrospun fiber fabrication. Likewise, needle-based electrospinning was used to fabricate electrospun shellac fibers loaded with KP extract to enhance the productivity of fibers with a small diameter.

Response surface methodology (RSM) with the correct design of trials is often used for formulation optimization. The Box–Behnken design (BBD) is a good RSM for studying formulation variables (independent factors) and their reciprocal influence on responses (dependent factors) [21,22]. The BBD, an independent quadratic model, places all of its design points either in the drawing’s center or evenly spaced from it on the cube’s ends [23]. It uses three levels of each factor, and all design points are inside the safe operating zone. Despite its limited coverage of the corners of nonlinear design space, the BBD is still considered the most powerful and proficient design [21,24,25]. Compared to three-level full-factorial design and central-composite design (CCD), it requires fewer experimental runs and is less exclusionary. This study optimized KP-extract electrospun fibers using a BBD.

To obtain the desired properties of electrospun fibers using a classical experimental design, the number of experiments required was relatively high due to the large number of parameters involved, which created time and cost constraints. To address these issues, a modern experimental design was introduced, which allowed for the planning of experiments that produced effective results with a small number of experimental trials [21,25]. KP extract-loaded electrospun shellac fibers were developed in this study, and the effects of independent variables, such as the shellac concentration, solution feed rate, electrical voltage, and extract concentration, were determined using an experimental design. The BBD was used to analyze the optimized values of each independent parameter. Nonetheless, the release kinetics of KP extract-loaded electrospun shellac fibers have not yet been documented. For this reason, the release kinetics of total methoxyflavones, including DMF, TMF, and PMF, were also examined in this work.

A bacterial infection may lengthen the time required for a wound to heal. The issue of antibiotic resistance has become an impediment to the treatment of skin infections. Consequently, the application of antibacterial substances derived from natural sources has received considerable attention. Previous research indicated that an ethanolic KP extract could inhibit growth of the bacteria *Staphylococcus aureus* [4] and *Escherichia coli* [26], which are responsible for skin infections. In this study, electrospun shellac fibers containing KP extract were tested for antibacterial activity.

## 2. Materials and Methods

### 2.1. Materials

Shellac flakes (Lot No. 55) with an average molecular weight of 1006 Da were purchased from Mahachai Shellac Company Limited, Samut Sakhon province, Thailand. KP rhizomes were kindly supported by Bangkok Lab & Cosmetic Co., Ltd., Ratchaburi province, Thailand. RCI Labscan Limited, Bangkok, Thailand, supplied 95% *v*/*v* ethanol (AR grade) and methanol (HPLC grade) for the preparation of the electrospun solution and mobile phase, respectively. The AR grade of acetic acid was purchased from Merck KGaA, Darmstadt, Germany. Ultrapure water was processed by a TKA Pacific-UP/UPW water purification system, Niederelbert, Germany. DMF, TMF, and PMF standards were procured from Indofine Chemical Company, Hillsborough, NJ, USA. *S. aureus* ATCC 6538P, *E. coli* DMST 4212, and *Pseudomonas aeruginosa* ATCC 9027 were the strains of bacteria used to test antibacterial activity. Tryptic soy agar and broth (TSA and TSB) purchased from HiMedia Laboratories Pvt. Ltd., Mumbai, India were utilized as growth media for the microorganisms. The positive control was powdered neomycin sulfate (Lot No. 18120199) supplied by Greater Pharma Co., Ltd., Bangkok, Thailand.

### 2.2. Preparation of Shellac Solutions Containing KP Extract

A fine powder of KP rhizomes was extracted with 95% *v*/*v* aqueous ethanol (solvent to solid ratio was 10 mL/g) in an ultrasonic bath at 25 °C for 1 h. The extract solution was filtered through a filter paper and then evaporated using a rotary evaporator (R-100, Buchi, Tokyo, Japan) under reduced pressure at 40 °C to achieve the KP extract. The yield of extraction was 4.13 ± 0.61%.

Shellac solutions with concentrations of 36–40% *w*/*w* were prepared by dissolving shellac flakes in 95% *v*/*v* aqueous ethanol at room temperature using a magnetic stirrer until completely dissolved. The KP extracts of varying concentrations were added to the shellac solutions according to the experimental design, and the desired volumes of the solutions were adjusted with 95% *v*/*v* ethanol. The obtained shellac solutions containing KP extract were subjected to an electrospinning process.

### 2.3. Electrospinning Process

The electrospinning process was carried out in ambient settings, which included a temperature control of 23–25 °C and a relative humidity of 40–60%. Each shellac solution containing KP extract was placed into a 10-mL syringe fitted with a metal needle and attached to a high-voltage power supply. Aluminum foil was employed as the collector, and the distance between the syringe needle tip and the collector was maintained at a constant 20 cm because of the restrictions of our electrospun machine. The spin collector was fixed at a constant site. However, our preliminary test (data not shown) revealed that 20 cm was the appropriate distance, which provided sufficient time for the solvent of the electrospun solution to vaporize and then dry. In addition, the nozzle diameter also played an important role in the fiber diameter and bead-to-fiber ratio. Nevertheless, based on our preliminary results, a nozzle with a low diameter showed the difficulty of the polymer ejection from the needle. Therefore, the fiber preparation with a small-diameter nozzle was not successful. In addition, a nozzle with a large diameter resulted in the accumulation of the polymer solution at the tip of the needle leading to non-continuous fiber production and a wide distribution of fiber diameter. Thus, the nozzle diameter was also fixed at 0.8 mm (Gauge No. 18). The fabrication parameters for the shellac fibers and shellac fibers incorporated with KP extract are shown in Table 1. All obtained fibers were kept in desiccators and subsequently characterized.

### 2.4. Experimental Design and Data Analysis

The BBD was taken to optimize the characteristics of the extract-loaded electrospun shellac fibers, including small diameter and bead-free fibers. To begin, screening of the significant parameters from the preliminary selection variables was determined. The second stage evaluated the optimal values of the significant parameters for the desired fiber qualities. This optimization design was performed to point out the values of each parameter that would result in the desired properties of the electrospun fibers. The independent parameters were analyzed at three different levels (−1 for the low level, 0 for the center level, and +1 for the high level). The independent parameters and their associated coded levels utilized for the BBD are listed in Table 1.

The combination of experiments consisted of 27 trials with each run being performed in triplicate. The model was fitted to a second-order polynomial equation in order to predict the optimal point of the BBD. The following Equation (1) denotes the quadratic model that was used to predict the responses:(1)$$Y=\beta 0+\sum_{i=1}^{4}\beta_{i}x_{i}+\sum_{i=1}^{4}\beta_{\mathrm{ii}}x_{i}^{2}+\sum_{i=1}^{4}\beta_{\mathrm{ij}}x_{i}x_{j}$$
where Y signifies the predicted response; β_0_ designates the intercept; β_i_, β_ii_, and β_ij_ refer to the regression coefficients of linear, quadratic, and interactive terms, respectively; and x_i_ and x_j_ represent the independent variables.

### 2.5. Model Validation

The obtained model was validated in order to ascertain its reliability. The triplicate experimental runs were conducted under the model-selected conditions. The experimental values were compared to the model-predicted values. The confidence level of 95% was used to compare the experimental and predicted values. Experiments were conducted in triplicate.

### 2.6. Morphological Characterization

A scanning electron microscope (SEM) (MIRA 3, Tescan, Brno, Czech Republic) was utilized to examine the morphology of the obtained fiber formulations. All samples were coated with gold and then fixed on the SEM stub. The images were photographed at an accelerating voltage of 10 kV with a working distance of 9 mm. After that, the fiber diameters and the bead-to-fiber ratios were calculated using JMicroVision version 1.2.7 software, Geneva, Switzerland. The bead-to-fiber ratio, also referred to as the bead population, was determined by counting the number of beads visible in a SEM photograph. Following that, the exact number of fibers was determined. The ratio of beads to fibers was determined by dividing the number of beads by the total number of fibers in the unit area of the SEM image [27].

### 2.7. Physicochemical Characterization of Fabricated Electrospinning Fibers

#### 2.7.1. Powder X-ray Diffraction (PXRD)

A powder X-ray diffractometer (MiniFlex II, Rigaku Corporation, Tokyo, Japan) was employed to ascertain the crystalline and/or amorphous behavior of the electrospun fibers. The diffractometer was set to 40 mV, 30 mA, and a scan range of 2θ from 5° to 40° at a speed of 4°/min with graphite monochromatized Cu Kα radiation (λ = 1.5406 Å).

#### 2.7.2. Differential Scanning Calorimeter (DSC)

The compatibility of each component and the thermal behavior of the formulated fibers were investigated by a differential scanning calorimeter (DSC 8000, Perkin Elmer, Rodgau, Germany). The selected-fiber formulations (accurately weigh approximately 2–3 mg) were individually sealed in aluminum pans and heated at a rate of 10 °C/min from 20 °C to 200 °C. Nitrogen gas was used as a carrier gas at a constant flow rate of 20 mL/min.

#### 2.7.3. Fourier Transform Infrared (FTIR) Spectroscopy

The molecular behavior and interaction between the KP extract and shellac matrix in the fabricated fibers were determined using a FTIR spectrometer (Nicolet Avatar 360, Ramsey, MN, USA). Each sample was triturated with potassium bromide (KBr), entirely blended in a mortar and pestle, compressed into a disc with a hydraulic press, and placed in a sample holder. The scanning process was carried out at a resolution of 4 cm^−1^ between 4000 and 400 cm^−1^. OMNIC FTIR software version 7.2a, Thermo Electron Scientific Instruments LLC, Madison, WI, USA was used to determine the FTIR spectra.

### 2.8. In Vitro Dissolution Study of KP Extract-Loaded Electrospun Shellac Fibers

According to the <724> drug release for transdermal systems and other dosage forms, United States Pharmacopeia 43-National Formulary 38 (USP 43–NF 38) [28], dissolution tests were conducted on the electrospun shellac fibers containing KP extract using a paddle-over-disk apparatus (USP Apparatus 5, Erweka GmBH, Heusenstamm, Germany). As a disk assembly component, the dried-fabricated fibers were packed into a 5-cm-diameter circular disk. The disk assembly was placed at the bottom of each mini dissolution vessel containing 200 mL of phosphate buffer solution (pH 7.4) as a dissolution medium. The experiment was conducted at 100 rpm in a 32 °C ± 0.5 °C temperature-controlled environment. At 20, 40, 60, 90, 120, 180, 240, 300, 360, 480, and 600 min, 3 mL of sample solution was taken out of each vessel and then replaced with fresh buffer solution. The concentrations of total methoxyflavones (DMF, TMF, and PMF), which are active components, were determined for each sample solution using the validated high-performance liquid chromatography with diode-array detection (HPLC-DAD) method. The peak area responses were converted to concentrations of methoxyflavones (DMF, TMF, and PMF) using linear equations of calibration curves, and then the cumulative percentage of total methoxyflavones released was computed. Each quantitative measurement was performed in triplicate. The release data obtained were fitted to mathematical kinetic models, mainly the zero-order kinetic model, first-order kinetic model, Higuchi model, and Korsmeyer–Peppas model, to comprehend the total methoxyflavone release mechanism and kinetics from the electrospun shellac fibers. Moreover, the mathematical analysis was conducted by computing the coefficient of determination (R^2^), Akaike information criterion (AIC), and model selection criterion (MSC) for the release data of methoxyflavones. These values were used to determine the most suitable release profiles for the samples.

### 2.9. Quantification of Methoxyflavones

An Agilent 1100 HPLC-DAD was used for the chromatographic analysis (Agilent Technologies, California, USA). On a Luna Omega Polar C18 column (5 μm, 100 Å, 4.6 mm × 250 mm) (Phenomenex Inc., Torrance, CA, USA), the separation runs were carried out at 40 °C. A mixture of aqueous acetic acid (0.1% *v*/*v*) and methanol (35:65) was used as a mobile phase. The injection volume was 20 μL, and the flow rate was 0.8 mL/min with isocratic elution. The detection UV wavelength was set at 254 nm.

The electrospun shellac fibers loaded with KP extract were successfully powdered. A 100 mg sample powder was precisely weighed into a 1.0 mL dry, clean volumetric flask. To aid in full solubility, the mobile phase was added and ultrasonically mixed. The same mobile phase was used to adjust the volume, and the analysis was performed using the 100 mg/mL sample solution. The validated method was then used to simultaneously measure the amounts of DMF, TMF, and PMF in various electrospun shellac fibers loaded with KP extract.

In the case of sample solutions obtained from an in vitro dissolution study of KP extract-loaded electrospun shellac fibers, each sample from the dissolution vessels was directly injected into the HPLC-DAD system, and the concentrations of methoxyflavones in each sample were calculated using their corresponding calibration curves.

### 2.10. Time-Kill Kinetics Assay

In this test, the pathogenic bacteria that cause wound infections, *S. aureus*, *E. coli*, and *Pseudomonas aeruginosa*, were used for the antibacterial assay. Each bacterium was transferred from the TSA to the TSB and incubated at 37 °C for 18–24 h. The absorbance at 600 nm of the bacterial suspensions was measured and compared to the 0.5 McFarland standard to estimate the number of bacteria (10^8^ CFU/mL; optical density = 0.05). Following this, the suspension cultures were diluted to 10^3^ CFU/mL. Before the test, the obtained fiber formulation was sterilized with UV light for 30 min. In the experiment, 100 mg of the electrospun shellac fibers with KP extract that were 5 × 5 cm^2^ in size were added to the medium cultures to make a final concentration of 10 mg/mL. The antibacterial standard, neomycin sulfate, at a concentration of 10 mg/mL was used as a positive control. The fiber devoid of KP extract served as a negative control. Throughout the experiment, all samples were stored in an incubator equipped with an automatic shaker. At each of the specified time intervals (15 min, 30 min, 1 h, 2 h, 4 h, and 6 h), 100 μL of each media sample was withdrawn and spread on an agar plate. All agar plates were incubated for 18–24 h at 37 °C. According to a previous study [29], the antimicrobial activity value (R) was calculated using the Equation (2):(2)$$R\;(\%)=\frac{B\;-\;C}{B}\;\times \;100$$
where B and C represent the microbial loads in CFU/mL for the initial point and each specified time point, respectively. The experiment was carried out in duplicate.

## 3. Results and Discussion

### 3.1. Optimization of Operating Conditions by BBD

#### 3.1.1. Statistical Analysis

A three-level BBD with 27 runs was used in this study including three replications at the center point. Appendix A contains the complete experimental design matrix used for optimization as well as the responses.

ANOVA results for the responses, namely fiber diameter and bead-to-fiber ratio, were analyzed according to Appendix A. When the models were analyzed, both models describing the responses were significant due to their low *p*-values and high F-values. Correspondingly, the values for lack-of-fit were not significant. Additionally, the three linear terms (x_1_, x_2_, and x_4_) have been identified as critical fiber diameter parameters. The coefficients of linear (x_1_ and x_4_) were considered significant terms in the bead-to-fiber ratio response. The R^2^ was used to evaluate the model’s goodness of fit. Both models had relatively high R^2^ values (0.7983 and 0.7652 for fiber diameter and bead-to-fiber ratio, respectively), indicating that they were practical and dependable.

#### 3.1.2. Graphical Representation of the Model and Response Surface Optimization

The three-dimensional (3D) response surfaces for fiber diameter and bead-to-fiber ratio were constructed using Design-Expert software version 8.0 as illustrated in Figure 3 and Figure 4. The interactive effects between the two parameters are demonstrated when the other parameters remain constant at the center of each experimental range.

For the fiber diameter analysis, the 3D response surfaces of fiber diameter are displayed in Figure 3. The results indicated that the most significant parameter was shellac concentration (x_1_) followed by extract concentration (x_4_) and solution feed rate (x_2_). As illustrated in Figure 3A,B, the diameter of the fibers increased with increasing shellac and extract concentrations. According to our previous study [13], increasing the concentration of shellac resulted in a significant increase in the chain entanglement of the shellac polymer and solution viscosity as well as a decrease in the stretching capability of the shellac solution. These factors contributed to the fibers’ increased diameter. As a result of this study, it was determined that a reduction in polymer concentration would result in a narrow fiber diameter; a similar result was reported by Xu et al. [30]. In terms of fiber diameter, the extract concentration followed a similar pattern to the shellac concentration, which was consistent with the previous report. With respect to the solution feed rate parameter, an increase in fiber diameter was also observed as the solution feed rate increased. This investigation established a strong correlation between the solution feed rate and the flow of polymer solution. An increased solution feed rate allowed for the rapid transfer of more polymer jets from the tip of the needle to the collector. As a result, the thicker fibers were produced on a continuous basis.

In the case of the bead-to-fiber ratio response, the shellac concentration was the most significant independent variable followed by the extract concentration (Figure 4). The remaining parameters, on the other hand, were not significant. As a consequence, the formation of beads was exclusively determined by the concentration of the shellac solution. The bead-to-fiber ratio decreased as the shellac solution concentration was increased as illustrated in Figure 4A,B. Increasing the shellac concentration resulted in increased polymer chain entanglements within the solution, which was required to maintain the jet’s continuity during electrospinning and avoid the formation of beads. When a low-viscosity shellac solution was used, the high concentration of solvent molecules and the lack of chain entanglements enhanced the surface-tension effect along the electrospinning jet, resulting in the formation of beads along the fiber. The findings of this study corroborate prior research [31,32].

Under the electrospinning conditions of 0.8 mL/h solution feed and 18 kV electrical voltage, the diameters of the fibers containing 1% *w*/*w* KP extract increased with increasing shellac concentrations from 36% *w*/*w* to 40% *w*/*w* (Figure 5A–C), whereas the diameters of the fibers containing 38% *w*/*w* shellac increased with increasing extract concentrations from 1% *w*/*w* to 5% *w*/*w* (Figure 5D–F). Furthermore, under the same electrospinning conditions, the bead-to-fiber ratios of the fibers containing 1% *w*/*w* KP extract decreased as the shellac concentrations increased from 36% *w*/*w* to 40% *w*/*w* (Figure 5A–C), whereas the bead-to-fiber ratios of the fibers containing 38% *w*/*w* shellac decreased as the extract concentrations increased from 1% *w*/*w* to 5% *w*/*w* (Figure 5D–F).

Following that, mathematical models were used to explain the relationship between the independent and response variables. The responses of fiber diameter and bead-to-fiber ratio are described in terms of coded variables, Equations (3) and (4), together with their corresponding R^2^ values of 0.7983 and 0.7652.
Fiber diameter = 744.07 + 200.66(x_1_) + 72.96(x_2_) + 124.97(x_4_)(3)
Bead-to-fiber ratio = 0.31 − 0.28(x_1_) − 0.07(x_4_)(4)

Additionally, the following Equations (5) and (6) express the actual (un-coded) factors:Fiber diameter = −3401.89 + 100.33(shellac concentration)
+ 182.39(solution feed rate) + 62.49(extract concentration)(5)
Bead-to-fiber-ratio = 5.66 − 0.14(shellac concentration) − 0.03(extract concentration)(6)

In summary, the small diameter of the fibers was achieved by using low shellac and extract concentrations in the electrospun solution and a slow solution feed rate. In turn, the high concentrations of shellac and extract resulted in the formation of bead-free fibers. As stated previously, the decreased shellac concentration led to a significant decrease in viscosity and a significant increase in electrical conductivity (data not shown), resulting in an increase in the electrospun solution’s stretching capacity. Consequently, fibers with a small diameter could be produced [19,33]. In contrast, an increase in the viscosity of the electrospun solution contributed to an increase in the diameter of the resulting fiber as a result of a reduction in the stretching capacity during the electrospinning process. In addition, high concentrations of shellac and KP extract resulted in a significantly high electrical conductivity accompanied by an increase in charge density and polymer jet solution instability during the electrospinning process leading to the formation of beaded and incomplete fibers [34]. The presence of beads on electrospun fibers is regarded as a flaw. The number of beads tended to decrease as the shellac and KP extract concentrations increased. A rise in the shellac’s viscosity could well have increased the interaction between the shellac polymer and solvent molecules. The possibility of producing beads decreased. Thus, the beaded fibers gradually transformed into smooth fibers [19].

In this investigation, a significant correlation was found between the solution feed rate and the fiber diameter. Due to the increased solution feed rate, more polymer jets were rapidly transferred from the needle tip to the collector resulting in an increase in fiber diameter. However, the solution feed rate had no effect on the bead-to-fiber ratio. The electrical voltage was deemed insignificant for both responses of solution feed rate and the bead-to-fiber ratio. According to a number of research articles, the diameter of the fiber expanded as the voltage was increased [35,36]. In contrast, a few research reports concluded that the fiber diameter decreased as the applied voltage increased [37,38]. Furthermore, others have reported that applied voltage has no effect on fiber diameter [39,40]. However, the limitations of this study could be attributed to the narrow voltage range of 12 kV to 24 kV. Low electrical voltage (<12 kV) impeded the formation of the fiber. At electrical voltages greater than 24 kV, an electric arc formed.

### 3.2. Optimization of Extract-Loaded Electrospun Shellac Fibers

Two criteria were used in this optimization method to achieve the desired ranges of fiber diameter and bead-to-fiber ratio. To establish the design space, an overlay plot, which is a two-dimensional contour plot, was used. The overlay depicts the range of possible response values. The yellow color signifies that the experimental data achieve the optimal area. The result in the gray-shaded area, on the other hand, indicates that the experimental data do not fit inside the optimization area [41]. The fiber diameter and bead-to-fiber ratio (the number of beads per fiber) were determined to be within the suitable ranges of 300–600 nm and 0–0.5, respectively, in our earlier work [13]. The overlay plot indicates that the optimal conditions for the production of KP extract-loaded electrospun shellac fibers with a smaller diameter (approximately 574 nm) and bead-free fiber (approximately 0.47) were as follows: shellac and extract concentrations of 37.25% *w*/*w* and 1.50% *w*/*w*, respectively. The other two parameters were 0.8 mL/h for solution feed and 18 kV for electrical voltage as illustrated in Figure 6.

### 3.3. Validation of the Established Models

Often, the validation study is determined at the completion of the experimental design process. Additional experiments were conducted in this work to analyze the validation processes for the two responses: fiber diameter and bead-to-fiber ratio. Appendix A summarizes the verification results. Under the specified conditions, which included a shellac concentration of 37.25% *w*/*w*, an extract concentration of 1.50% *w*/*w*, a solution feed rate of 0.8 mL/h, and an electrical voltage of 18 kV, the predicted diameter and bead amount were highly correlated with the experimental data. These findings established that the study’s developed models were applicable and dependable.

### 3.4. Physicochemical Characterization of Fabricated Electrospun Fibers

PXRD, DSC, and FTIR were used to characterize the physicochemical properties of the KP extract-loaded electrospun shellac fibers, including crystalline characteristic, thermal behavior, and extract–polymer interaction. The optimized formula with a small diameter and lower number of beads was chosen to represent all fibers that will be further characterized.

#### 3.4.1. Powder X-ray Diffractometry (PXRD)

The PXRD diffractograms of the shellac flakes, KP extract, their physical mixture, electrospun shellac fibers, and electrospun shellac fibers loaded with an optimized formula of KP extract are shown in Figure 7.

The shellac flakes and KP extract did not show a clear diffraction peak, which suggested the amorphous characteristic of both compounds. After electrospinning, the shellac fibers and KP extract-loaded electrospun shellac fibers (optimized formula) also demonstrated the same diffractogram as observed in the shellac and KP extract. The result indicated that electrospinning did not interfere with the molecular packing or crystalline characteristics, even though the physical appearance was significantly changed under high voltage. Nevertheless, it was noted that the shellac flakes or shellac fibers did not show a complete halo pattern; several peaks, including a broad diffraction peak around 2θ = 9.4° and a small diffraction peak at 2θ = 18.2°, were observed, while the KP extract also demonstrated small diffraction peaks at 2θ = 11.5°, 14.7°, and 25.4°. In this case, it might be possible that the shellac and KP extract possess a partial crystalline characteristic that needs to be further confirmed by calculating the degree of crystallinity and investigated by other instrumental analyses including DSC [42].

The degree of structural organization in a solid is referred to as “crystallinity”. Atoms or molecules are arranged in a crystal in a consistent and repeating manner. Due to the fact that polymers do not have a uniform molecular weight, they typically only construct partially crystalline structures or semi-crystalline structures [43]. It is generally known that a number of physical techniques, including X-ray diffraction, calorimetry, density estimation, infrared spectroscopy, and nuclear magnetic resonance, can be used to quantify the degree of crystallinity. It has been asserted that X-ray diffraction is fundamentally superior to other techniques for determining crystallinity [44]. By integrating peak areas under the PXRD diffractograms using Origin software (OriginLab Corporation, Northampton, MA, USA), the degree of crystallinity for the shellac flakes, KP extract, physical mixture of shellac flakes and KP extract, electrospun shellac fibers, and KP extract-loaded electrospun shellac fibers with an optimized formula (Table 2) was determined. When comparing the shellac flakes with the electrospun shellac fibers, the percent crystallinity of both materials was not clearly different. Additionally, the percent crystallinity of the optimized KP extract-loaded electrospun shellac fibers was also in a similar region as compared to that of the physical mixture of shellac flakes and KP extract. The finding indicated that electrospinning did not affect crystal packing or recrystallization, although rapid evaporation of solvent occurred during the process.

#### 3.4.2. Differential Scanning Calorimetry (DSC)

Figure 8 illustrates the DSC thermograms of the shellac flakes, KP extract, physical mixture of shellac flakes and KP extract, electrospun shellac fibers, and electrospun shellac fibers loaded with KP extract. The thermograms of the shellac flakes and electrospun shellac fibers revealed endothermic peaks around 60 °C corresponding to their melting points as previously reported [45]. In order to investigate the endothermic peaks observed, we also investigated the shellac samples by hot-stage microscopy (Appendix A). Both the shellac flakes and fibers showed the transformation from solid to liquid droplet after increasing the heating temperature over 65 °C, confirming that endothermic peaks were caused by the melting of the shellac samples. According to the results of PXRD of the shellac flakes and shellac fibers, the melting behavior might arise from the partial crystalline characteristic of the shellac. Additionally, the broad endothermic peak was also caused by the adsorbed water, which corresponded with the moisture content in the shellac sample.

Meanwhile, the KP extract revealed endothermic peaks at 126.05, 133.68, and 149.32 °C, which corresponded to the melting points of the methoxyflavone compounds in the KP extract. Nevertheless, the evaporation and degradation of KP could also be the cause of the endothermic peaks. Therefore, we further investigated the KP extract by thermogravimetric analysis (TGA). A slight weight loss was observed at temperatures below 200 °C, while a dramatic weight loss was observed after 250 °C, which suggested that the endothermic peaks below were also probably due to the evaporation of some volatile contents but not to the degradation. Appendix A show TGA thermograms of KP extract and samples, respectively.

According to a previous study [46], the thermograms with a broad peak of the physical mixture of shellac flakes and KP extract and the KP extract-loaded electrospun shellac fibers were similar to those of the shellac flakes and electrospun shellac fibers, respectively, presumably because the physical mixture and the electrospun shellac fibers loaded with KP extract contained significantly more shellac than extract. The electrospun shellac fibers loaded with KP extract, on the other hand, exhibited a different endothermic peak as compared with the physical mixture of shellac flakes and KP extract. The identification and measurement of extremely low-energy transitions of a few micrograms of KP extract in shellac fibers might not even be possible at standard DSC scan rates. In order to confirm the interaction between methoxyflavones and shellac molecules and the dispersion of KP extract in shellac fibers, FTIR, as described in the following section, was utilized.

Additionally, the latent heat (enthalpy) can be calculated from the area under the DSC thermogram’s curve. The melting enthalpies (ΔH_m_, J/g) of the physical mixture and the KP extract-loaded electrospun shellac were 34.98 and 18.97 J/g, respectively. The results suggested that the KP extract might be dispersed in the shellac matrix and resulted in the lowering of the melting enthalpy of shellac.

#### 3.4.3. Fourier Transform Infrared (FTIR) Spectroscopy

FTIR spectroscopy was used to investigate the structural changes that occurred during electrospinning as well as interactions between shellac and KP extract in the electrospun shellac fibers loaded with KP extract. As illustrated in Figure 9, the spectra of the shellac flakes and electrospun shellac fibers contained broad peaks at 3448 and 3422 cm^−1^, respectively, corresponding to O-H stretching. Meanwhile, the peaks at 1718 and 1254 cm^−1^ corresponded to C=O and C-O stretching, respectively. Moreover, the alkyl C-H stretching at 2928 or 2927 cm^−1^ and 2854 cm^−1^ as well as the alkenyl C=C vibration peak at 1637 or 1639 cm^−1^ were detected [47]. According to an earlier report [7], in the case of methoxyflavones in KP extract, the characteristic peaks of C=C stretching at 1638 and 1604 cm^−1^ and C-H vibration at 2926 and 2851 cm^−1^ were apparently observed to be the major peaks on the spectrum. Not only were the characteristic peaks of the physical mixture similar to those of shellac, but also a small peak of KP extract at 1607 cm^−1^ (C=C stretching) was detected. Additionally, the intensities of all peaks on the spectrum of the physical mixture tended to be lower than those of the individual spectra. The decrease in peak intensities on the spectrum of the physical mixture could be due to the substantially greater amount of shellac than KP extract. The FTIR spectrum of the KP extract-loaded electrospun shellac fibers with an optimized formula had a similar appearance to that of the physical mixture, but the intensities of the shellac peaks were increased, and the C=C stretching peak of the KP extract at 1607 cm^−1^ was completely lost. In methoxyflavones, the peak at 1607 cm^−1^ was attributed to in-plane vibrations of the benzene ring containing a carbonyl chromophore [7]. The intensity of this peak was also affected by the bonding during the electrospinning process. In the shellac polymer, the peaks at 1718 cm^−1^ corresponding to C=O stretching and the intensity of this peak were affected by the elongation of polymers during the electrospinning process, so a higher intensity could be observed [48]. Additionally, the alkane -CH_2_ bending characteristic of the shellac’s -CH_2_OH was shifted from 1466 to 1460 cm^−1^ in the electrospun shellac fibers loaded with KP extract due to intermolecular bonding at the -CH_2_OH side chain of the shellac molecule. Thus, a lost peak at 1607 cm^−1^ for the KP extract; increased intensities for all peaks, particularly the peak at 1718 cm^−1^ for the shellac polymer; and a shift of the -CH_2_ bending for the shellac’s -CH_2_OH could be interpreted as evidence of the KP extract dispersion within the shellac matrix in the electrospun fibers. The results indicated that the hydrogen bonding between the hydroxyl groups at -CH_2_OH in the shellac and the carbonyl groups at C-4 in the methoxyflavones might just have occurred.

### 3.5. Determination of Methoxyflavone Content and Loading Capacity

Important fiber preparation parameters include total methoxyflavone percentage and loading capacity. The weight of total methoxyflavone content analyzed by HPLC-DAD relative to the weight of the fiber is referred to as loading capacity [49]. As shown in Table 3, samples No. 1–4 were produced under varying conditions, including KP extract concentration, shellac concentration, solution feed rate, and electrical voltage. According to Equations (7) and (8), the % content of total methoxyflavones and loading capacity were calculated, respectively.
(7)$$\%\;\mathrm{Content}\;\mathrm{of}\;\mathrm{total}\;\mathrm{methoxyflavones}=\frac{\mathrm{Actual}\;\mathrm{concentration}\;\mathrm{of}\;\mathrm{methoxyflavones}\;\mathrm{in}\;\mathrm{fibers}\;(\mathrm{mg}/g\;\mathrm{fiber})}{\mathrm{Theoretical}\;\mathrm{concentration}\;\mathrm{of}\;\mathrm{methoxyflavones}\;\mathrm{in}\;\mathrm{fibers}\;(\mathrm{mg}/g\;\mathrm{fiber})}\;\times \;100$$
(8)$$\mathrm{Loading}\;\mathrm{capacity}=\frac{\mathrm{Total}\;\mathrm{methoxyflavone}\;\mathrm{content}\;\mathrm{incorporated}\;\mathrm{in}\;\mathrm{fibers}}{\mathrm{Weight}\;\mathrm{of}\;\mathrm{fibers}}\;$$

According to the results (Table 3), the % contents of total methoxyflavones ranged between 94.88 ± 2.72% and 113.33 ± 6.71%, and the % loading capacity ranged between 0.33 ± 0.00% and 0.88 ± 0.01%. According to the results, the total methoxyflavones could be loaded more when the voltage was raised. This is due to the fact that shellac fibers have a greater ratio of surface area to volume, which could increase their capacity for loading methoxyflavones [50]. The behavior of methoxyflavones when exposed to a high electrical voltage should be investigated further.

### 3.6. In Vitro Methoxyflavone Release Behavior

According to the results of ANOVA, the shellac and KP extract concentrations had the greatest impact on the fiber diameter and bead-to-fiber ratio responses. Certain formulations were chosen for the in vitro release of methoxyflavones in an effort to reduce the number of experimental runs required to determine the release profile. This section’s findings could explain the correlation between the responses (fiber diameter and bead-to-fiber ratio) and the release profile. The fiber formulations whose release profiles were examined were divided into two groups: (1) formulations with 3% *w*/*w* KP extract concentration and various shellac concentrations, such as 36% *w*/*w* shellac, 38% *w*/*w* shellac, and 40% *w*/*w* shellac formulas; and (2) formulations with 40% *w*/*w* shellac concentration and various KP extract concentrations, such as 1% *w*/*w* KP extract, 3% *w*/*w* KP extract, and 5% *w*/*w* KP extract formulas. The optimized formula contained shellac and KP extract concentrations of 37.25% *w*/*w* and 1.50% *w*/*w*, respectively. All formulations were fabricated under optimal conditions with a solution feed rate of 0.8–1.2 mL/h and an electrical voltage of 18–24 kV. The in vitro release profiles of total methoxyflavones from the intact KP extract and KP extract-loaded electrospun shellac fibers with different formulas in phosphate buffer solution (pH 7.4) at 32 °C ± 0.5 °C are depicted in Figure 10. In comparison to intact KP extract, all formulations of KP extract-loaded shellac fibers showed a considerable improvement in the dissolution rate. Furthermore, all fabricated electrospun fibers displayed a similar pattern of release. As the amount of shellac polymer increased or as the percentage of extract dropped, the cumulative percentage of total methoxyflavone release increased.

As shown in Figure 10, the initial release occurs within the first 3 to 4 h followed by a slow and steady release for up to 10 h. Total methoxyflavones were gradually liberated from the shellac-based electrospun fibers according to the findings. Within the first 3 h, 36% *w*/*w* shellac, 38% *w*/*w* shellac, 40% *w*/*w* shellac, and 1% *w*/*w* KP extract fibers released approximately 80% of the total methoxyflavones, whereas 3% *w*/*w* KP extract and 5% *w*/*w* KP extract fibers released only 40–60% of the total methoxyflavones. When the concentration of the KP extract was maintained at a constant concentration, the formulations with a higher concentration of shellac polymer resulted in a slower and longer release. As mentioned in the previous section, FTIR (Figure 9) could confirm the hydrogen bonding between the hydroxyl groups at -CH_2_OH in the shellac molecules and the carbonyl group at C-4 in the methoxyflavones due to a lost peak of C=C stretching of the KP extract at 1607 cm^−1^, an increment of peak intensity of C=O stretching of shellac at 1718 cm^−1^, and a shift of the -CH_2_ bending of -CH_2_OH in the shellac. According to the findings, a higher proportion of shellac induces more cross-linking with methoxyflavones, which alters the total methoxyflavone release profiles from the KP extract-loaded electrospun shellac fibers. Moreover, the slower release was also observed in the fibers with a greater concentration of KP extract and constant shellac concentration, particularly for the formulation containing 5% *w*/*w* KP extract. This could be explained by the fact that a higher extract concentration led to a decreased specific surface area of fibers (larger-diameter fibers), indicating a slower fiber release rate [51].

To determine the most appropriate release model for the samples, the statistic values, mainly R^2^, AIC, and MSC, were considered. The best model can be determined by taking the R^2^ that is highest and nearest to 1.0, the lowest AIC value, and the highest MSC value [52]. The release pattern of the KP extract-loaded electrospun shellac fibers fitted best to Korsmeyer–Peppas as shown in Appendix A with the R^2^ values (0.8995–0.9972) being greater than those of the other mathematical models together with the lowest AIC values (13.2558–28.2652) and the highest MSC values (1.1850–4.9303). Furthermore, all formulations had a release exponent (*n*) of 0.554–0.747, which was greater than 0.50 but less than 1.00 indicating non-Fickian diffusion (anomalous transport). Appendix A show a mathematical model that fits the KP extract release profiles of KP extract-loaded electrospun shellac fibers containing shellac and KP extract at various concentrations. The results were the same as those found by Rezaei and Nasirpour [53]. The release of methoxyflavones might be governed by the disentanglement and erosion of shellac. These findings suggested that KP extract-loaded electrospun shellac fibers could be used as a medication delivery system with controlled release.

The release profile provides detailed information about the involved mass-transport mechanism. Numerous mathematical models were indispensable for predicting the release patterns. To combat pathogens entering the human body, zero-order kinetics are typically required. The explanation could be attributed to the fact that zero-order kinetics provide a rapid release of active substances, a crucial factor in preventing pathogen invasion [54]. However, all formulations in this study exhibited the Korsmeyer–Peppas pattern, indicating the previously described polymeric matrix.

### 3.7. Time-Kill Kinetics Assay

A time-kill kinetics study was performed to investigate the pharmacodynamic profile of KP extract-loaded shellac fibers against gram-positive *S. aureus*, gram-negative *E. coli*, and gram-negative aerobic-facultatively anaerobic *P. aeruginosa* bacteria. In this test, the optimized KP extract-loaded shellac fibers with 37.25% *w*/*w* shellac and 1.50% *w*/*w* KP extract were used. Appendix A depicts the microbial loads of *S. aureus*, *E. coli*, and *P. aeruginosa* over a 6-h period. The number of *S. aureus* exposed to the KP extract-loaded shellac fibers decreased more gradually. The release of antibacterial methoxyflavones from the shellac electrospun fibers was nevertheless dependent on the solubility of the shellac fibers in the test medium. The results indicated that the antibacterial activity of the shellac fibers loaded with KP extract tended to be relatively long-lasting. Thus, it seemed that KP extract-loaded shellac fibers could be used as antibacterial wound dressings against *S. aureus* for 6 h (88.07% of inhibition).

As determined by *E. coli* susceptibility testing, *E. coli* was more resistant to the KP extract-loaded shellac fibers with 9.17% and 21.10% of inhibition over a 4-h and 6-h period, respectively. However, the shellac fibers loaded with KP extract lacked antibacterial activity against *P. aeruginosa*; the number of this bacterium increased dramatically with time (Appendix A). Furthermore, neomycin sulfate, a standard drug, had a bactericidal effect, inhibiting the growth of all tested strains within 15 min.

This study revealed that the KP extract-loaded shellac fibers were less effective against *E. coli* than *S. aureus*, suggesting a difference in the polarity of the outer membranes of gram-positive and gram-negative bacteria. It is known that the hydrophilic outer membrane of gram-negative bacteria is resistant to numerous antibacterial agents. Consequently, gram-negative bacteria are typically less susceptible to antibacterial agents than gram-positive bacteria [55].

Chronic and difficult-to-heal wounds affect health, social, and economic outcomes worldwide. Wound healing requires numerous biological and molecular mechanisms to regenerate tissue. Coagulation, inflammation, and clearance of damaged matrix components are the most critical physiological processes followed by cellular proliferation and migration, angiogenesis, matrix synthesis and deposition, re-epithelialization, and remodeling [56]. However, the antibacterial activity of the KP extract-loaded electrospun shellac fibers was not particularly noteworthy. Our past research showed that the KP extract added to these fibers had anti-inflammatory and antioxidant properties [5]. Both effects might very well accelerate wound healing. The methoxyflavones found in the ethanolic KP extract enhanced the antibacterial activity of gentamicin to combat bacterial resistance against *P. aeruginosa*, *Klebsiella pneumoniae*, and *Acinetobacter baumannii* strains resistant to carbapenems [57]. Therefore, KP extract should be combined with antibiotics to create electrospun shellac fibers for more efficient wound healing.

The KP extract-loaded electrospun shellac fibers exhibited antibacterial activity against a variety of bacteria, particularly gram-positive bacteria, and could be of great use in enhancing the antibacterial activity of antibiotics. Due to the critical issue of drug resistance, it is becoming increasingly difficult to select the most effective treatment for wound healing. Natural product-based treatments are becoming more and more popular [58,59]. In the past, they were used to treat wounds and kill bacteria that were resistant to multiple drugs [58,59]. In prior research, the combination of drug-delivery systems and effective components of herbal extract produced a synergistic antibacterial effect [60]]. In addition to maximizing the antibacterial activity of herbal extracts, the electrospun shellac fibers also had a high level of safety in comparison to other treatments.

## 4. Conclusions

In order to optimize the fiber diameter and bead-to-fiber ratio, it was possible to produce KP extract-loaded electrospun shellac fibers in this study. The shellac concentration had the most significant effect on both responses followed by the extract concentration. The solution feed rate was also a significant factor in fiber diameter response. However, there was no effect of the solution feed rate on the bead-to-fiber ratio. In both responses, the electrical voltage was deemed insignificant. In summary, integrating optimization with BBD would achieve effective results with a small number of experiments. This release mechanism analysis found that the Korsmeyer–Peppas model suited the release data with R^2^ values from 0.8995 to 0.9972. Under neutral pH circumstances, the electrospun shellac fibers facilitated the release of the KP extract. Therefore, KP extract-loaded electrospun shellac fibers could be developed for transdermal delivery systems with a controlled release of bioactive herbal extract. Based on the results of the time-kill kinetics assay, the optimized KP extract-loaded shellac fibers exhibited sustained antibacterial activity against *S. aureus*. Consequently, it is possible to develop the optimized fibers into wound dressings.

## Figures and Tables

**Figure 1 pharmaceutics-15-00123-f001:**
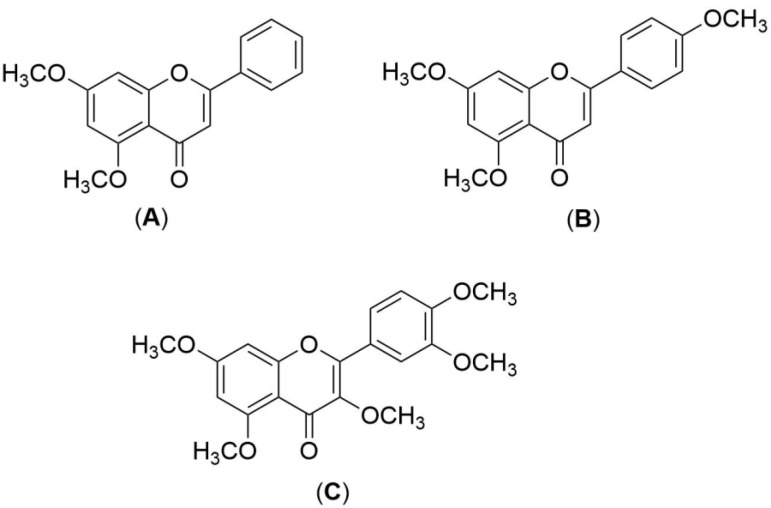
Chemical structures of (**A**) 5,7-dimethoxyflavone (DMF), (**B**) 5,7,4′-trimethoxyflavone (TMF), and (**C**) 3,5,7,3′,4′-pentamethoxyflavone (PMF).

**Figure 2 pharmaceutics-15-00123-f002:**
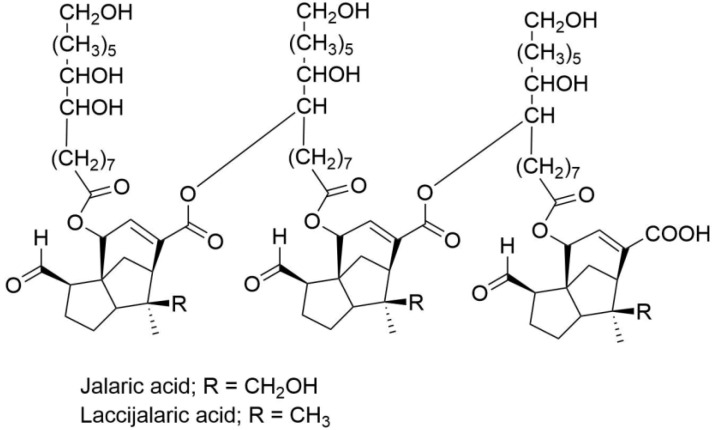
Chemical structure of shellac.

**Figure 3 pharmaceutics-15-00123-f003:**
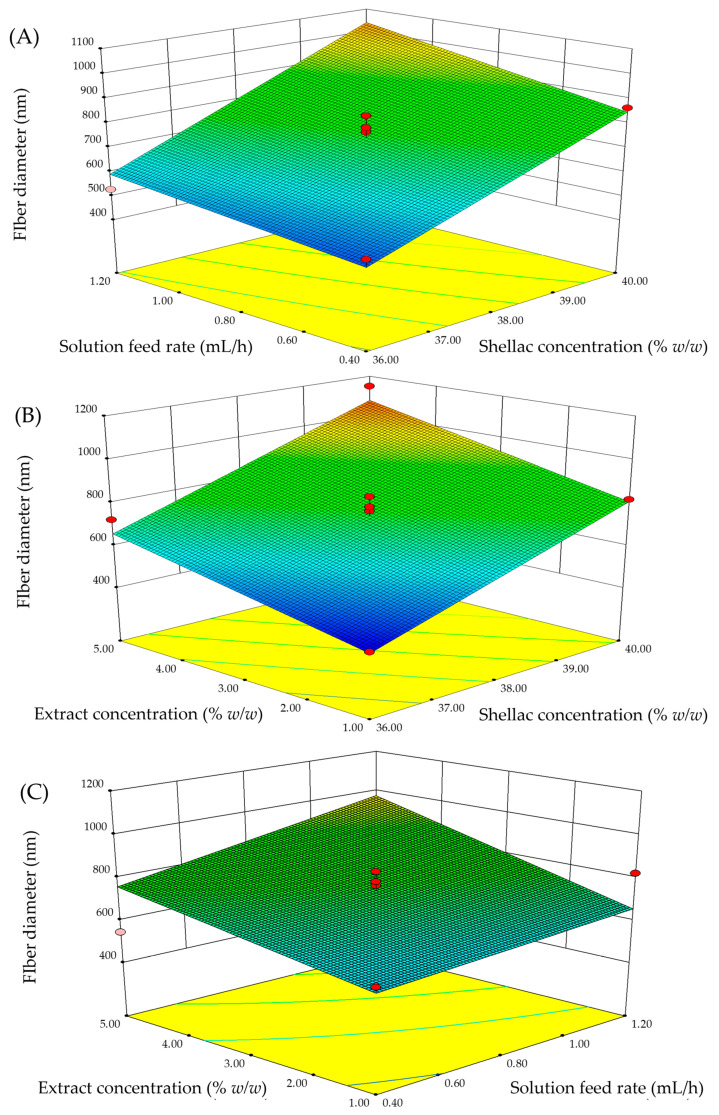
3D response surface plots with different parameters for fiber diameter: (**A**) shellac concentration and solution feed rate, (**B**) shellac concentration and extract concentration, and (**C**) extract concentration and solution feed rate.

**Figure 4 pharmaceutics-15-00123-f004:**
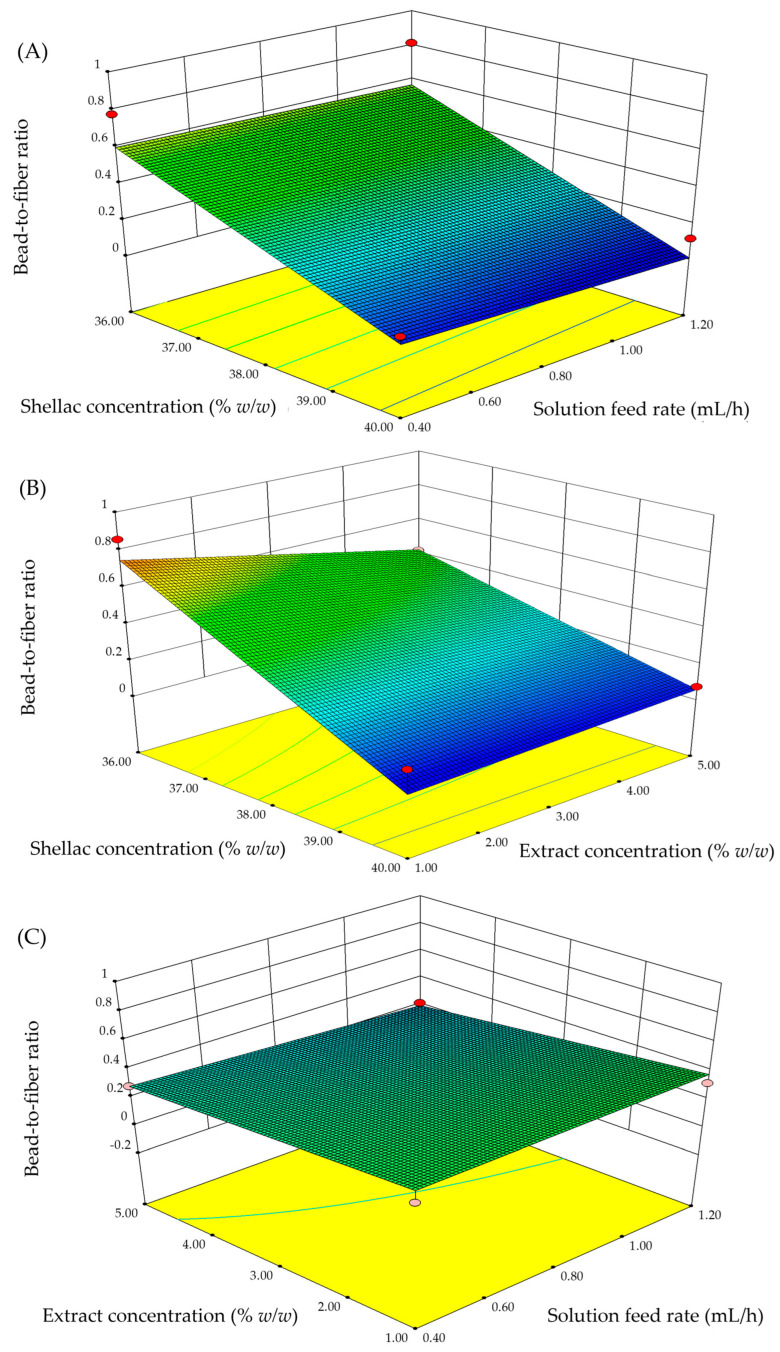
3D response surface plots with different parameters for bead-to-fiber ratio: (**A**) shellac concentration and solution feed rate, (**B**) shellac concentration and extract concentration, and (**C**) extract concentration and solution feed rate.

**Figure 5 pharmaceutics-15-00123-f005:**
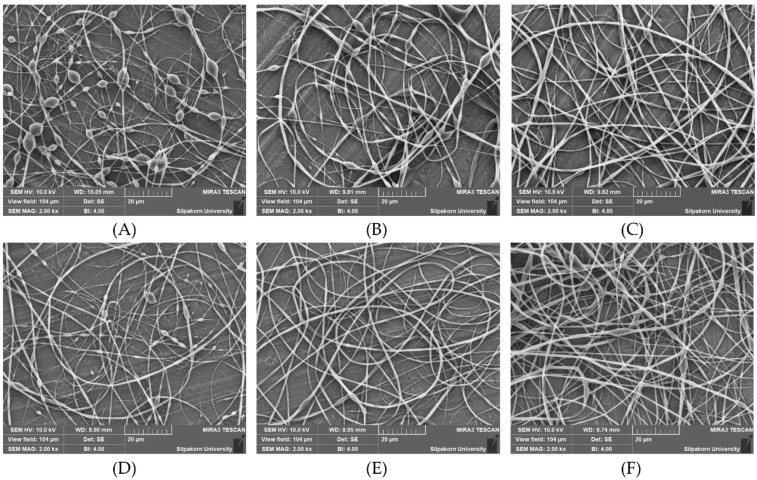
SEM images of fibers with 1% *w*/*w* extract and varying shellac concentrations of (**A**) 36% *w*/*w*, (**B**) 38% *w*/*w*, and (**C**) 40% *w*/*w* as well as fibers with 38% *w*/*w* shellac and varying extract concentrations of (**D**) 1% *w*/*w*, (**E**) 3% *w*/*w*, and (**F**) 5% *w*/*w* obtained under electrospinning conditions of 0.8 mL/h solution feed and 18 kV electrical voltage.

**Figure 6 pharmaceutics-15-00123-f006:**
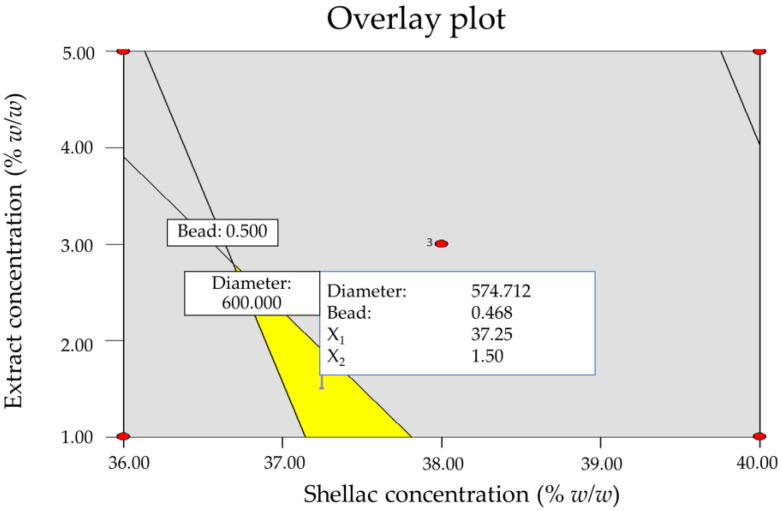
An overlay plot for the optimization of KP extract-loaded electrospun shellac fibers.

**Figure 7 pharmaceutics-15-00123-f007:**
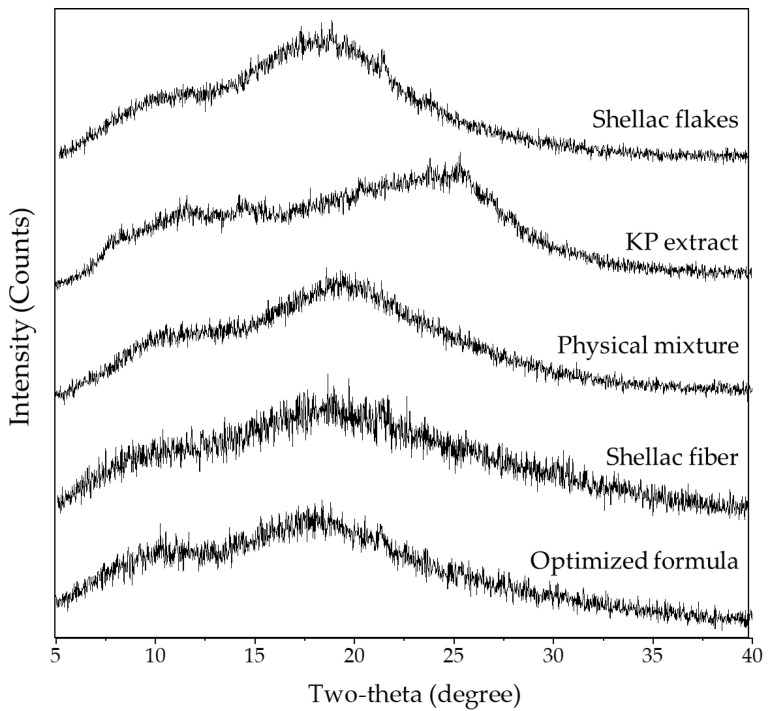
PXRD diffractograms of shellac flakes, KP extract, physical mixture of shellac flakes and KP extract, electrospun shellac fibers, and KP extract-loaded electrospun shellac fibers with an optimized formula.

**Figure 8 pharmaceutics-15-00123-f008:**
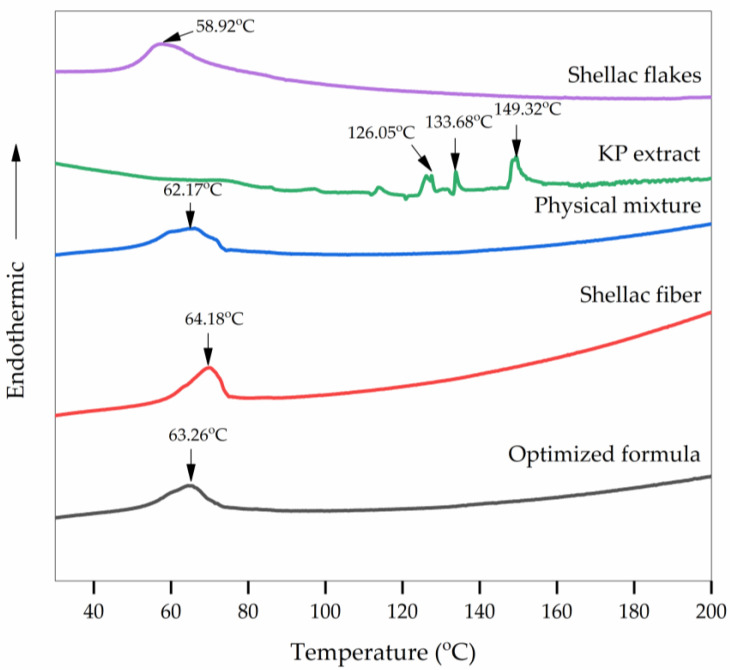
DSC thermograms of shellac flakes, KP extract, physical mixture of shellac flakes and KP extract, electrospun shellac fibers, and KP extract-loaded electrospun shellac fibers with an optimized formula.

**Figure 9 pharmaceutics-15-00123-f009:**
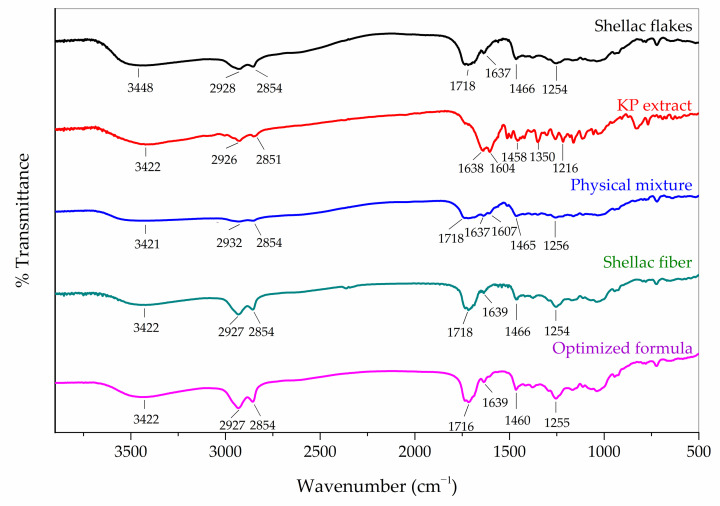
FTIR spectra of shellac flakes, KP extract, physical mixture of shellac flakes and KP extract, electrospun shellac fibers, and KP extract-loaded electrospun shellac fibers with an optimized formula.

**Figure 10 pharmaceutics-15-00123-f010:**
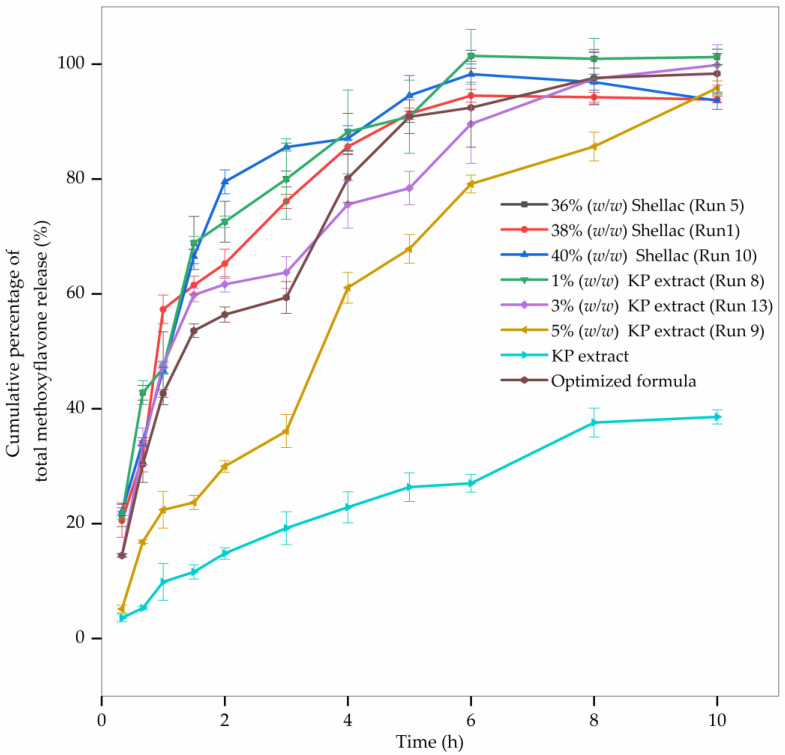
In vitro release profiles of total methoxyflavones from the intact KP extract and KP extract-loaded electrospun shellac fibers with different formulas in phosphate buffer solution (pH 7.4).

**Table 1 pharmaceutics-15-00123-t001:** Independent parameters and their corresponding levels in the BBD for the optimization process.

Independent Parameters	Symbol	Corresponding Levels		
		Low Level (−1)	Center Level (0)	High Level (+1)
Shellac concentration (% *w/w*)	x_1_	36	38	40
Solution feed rate (mL/h)	x_2_	0.4	0.8	1.2
Electrical voltage (kV)	x_3_	12	18	24
KP extract content (% *w/w*)	x_4_	1	3	5

**Table 2 pharmaceutics-15-00123-t002:** Degree of crystallinity.

Sample	Degree of Crystallinity (%)
Shellac flakes	4.13 ± 0.08 ^A^
KP extract	4.65 ± 0.07 ^A^
Physical mixture of shellac flakes and KP extract	4.10 ± 0.64 ^A^
Electrospun shellac fibers	4.03 ± 0.54 ^A^
Optimized KP extract-loaded electrospun shellac fibers	4.06 ± 0.55 ^A^

Mean ± SD, *n* = 3. There are statistically significant differences (*p* < 0.05) between values in the same column with different superscript characters. The identical character signifies no significant distinction.

**Table 3 pharmaceutics-15-00123-t003:** Assay results of methoxyflavones (DMF, TMF, and PMF) and loading capacity in KP extract-loaded electrospun shellac fibers.

Sample No.	Shellac Concentration (% *w/w*)	Electrical Voltage (kV)	Methoxyflavones	% Content of Each Methoxyflavone	% Content of Total Methoxyflavones	Loading Capacity
1	36	12	DMF	105.32 ± 6.01	113.33 ± 6.71 ^B^	0.35 ± 0.02 ^A^
			TMF	109.14 ± 6.70		
			PMF	126.31 ± 7.47		
2	36	24	DMF	85.20 ± 2.85	94.88 ± 2.72 ^A^	0.86 ± 0.02 ^B^
			TMF	94.64 ± 1.96		
			PMF	105.53 ± 3.82		
3	38	12	DMF	95.37 ± 1.04	106.17 ± 1.41 ^B^	0.33 ± 0.00 ^A^
			TMF	111.15 ± 6.18		
			PMF	112.50 ± 1.10		
4	38	24	DMF	87.25 ± 1.30	97.36 ± 1.27 ^A^	0.88 ± 0.01 ^B^
			TMF	98.48 ± 2.17		
			PMF	107.27 ± 2.19		

Mean ± SD, *n* = 3. Significant differences exist between values in the same column with different superscript characters (*p* < 0.05).

## Data Availability

The datasets used and/or analyzed during this study are available upon reasonable request from the corresponding author.

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
