# Peer review of "Fabrication, Optimization, and Characterization of Antibacterial Electrospun Shellac Fibers Loaded with Kaempferia parviflora Extract"

_pharmaceutics, 2022, doi:10.3390/pharmaceutics15010123_

Round 1

Reviewer 1 Report

This work could ultimately be of interest to the readership of Pharmaceutics. However, there are a number of problems with the MS as it stands which need to be addressed before publication can be considered. I set these out below, and recommend the MS be reconsidered after major revisions.

My main problem with the paper is that it appears to me to be two papers bolted together, with minimal connection between the components. The HPLC-DAD method does not seem to me very novel, and in my view should be confined to the methods/supporting information and not discussed in the results/title/abstract of the paper. This would help improve the focus of the paper, and also reduce the length - 30 pages for a research manuscript is really too much.

Other, more minor, issues are as follows:

>> Abstract
- Please pay more attention to numerical accuracy. It is not meaningful to quote diameters to 0.01 nm accuracy.
- What is "0.48" refering to? Needs to be explained.

>> Introduction
- It is stated that "Electrospinning was a widely applicable technique for increasing the perme-ability of bioactive compounds through the skin, and a variety of low solubility active compounds could be loaded into the fibers for controlled release", but no references are cited to support this statement.
- P4, para 1 - it is stated that a goal of the study is to develop a method of quantifying the flavones in the fibres. But this is not accurate - the fibres need to be dissolved or the flavones extracted from them before quantification can be undertaken.
- I do not see the novelty of the HPLC-DAD method. This is a standard analytical technique.

>> Methods
- Please give the MW of the shallac used.
- What is the other 5% in the 95% v/v ethanol solvent system?
- Table 1 - the authors quote voltages of 0.4 - 1.2 kV, but this is very low. Do they mean kV/cm?
- Section 2.7.1 - please give the wavelength of the X-rays used.
- Section 2.10 - what mass/area of fibre mat was used in the antibacterial experiments?

>> Results/Discussion
- Table 2- fibre diameter should be rounded off to nearest nm. Please quote values for diameter/bead-to-fibre ratio as mean +/- SD.
- I think Tables 2 and 3 could usefully be moved to supporting information to make the manuscript more succinct.
- Table 4 - same comments as for Table 2.
- The author state that the XRD data show their frmulations to be amorphous, but then quote melting points from the DSC data. An amorphous material cannot melt! These observations are thus contradictory, which needs to be discussed and explained. Are the peaks at 50-63 C really melting? I don't think they are.....
- Fig 8 - please mark exo/endo direction.
- The IR discussion mentions "the type of polymorphic form" of the KP extract. This is again contradictory to the XRD data, which show the formulation to be amorphous. An amorphous form is not "polymorphic". Please revise text for clarity.
- What is the encapsulation efficiency and drug loading in the optimised fibres? Please remove the later HPLC-DAD section, and instead simply quote the EE% and DL% here.
- Fig 10 - error bars are needed on this plot, and the number of replicate experiments needs to be stated. A number of different formulations are shown here, but there are no characterising data presented on them above. All formulations discussed in the paper need to be characterised in full.
- What release profile do we want/need to see for the desired application? Please discuss this.
- Please include the plots showing the fit of the Peppas model in supportng information. R2 values alone can be misleading, and the fits themselves need to be provded too.
- Section 3.6 - please remove or move to supporting information. This disrupts the flow of the MS, and makes it too long. Table 6 and Fig 11 should also be moved. The HPLC-DAD method is presumably used in the release studies, so it makes no sense to discuss it after the release data!
- Fig 11 is not of publication qualty - axis labels are not legible, and the resolution is very poor. Please revise and include a new image, as supporting information (NOT in the main MS).
- Section 3.8 - are the fibres used here the same as the optimal fibres discussed above?
- Fig 12 needs error bars.
- In general, the fibres seem to have very little effect on the bacteria, even in the case of S Aureus. Is this expected? WHat do I need to have to get effective wound healing?

Reviewer 2 Report

In this manuscript authors develop a Kaempferia parviflora extract based on electrospun fibers for transporting methoxyflavones. Further authors used a Box-Behnken design to determine the optimal production parameters. This manuscript needs major revision before it could be considered to the next level.

Title needs to change preciously at present it is so long and lack of flow of reading.

No line numbers.

All the tables and figures are not meet the criteria of scientifically. Please modified all the tables and figures format.

Authors need to describe the importance of BBD in the manuscript. Cite this manuscript IET Nanobiotechnology 14 (2020) 654-661.

Table 1 caption seems to have a different font style than other text.

Section 2.6, analysis condition needs to be mentioned. Such as working distance, current, voltage, etc.

Section 2.7.3, Fourier transform infrared spectroscopy (FTIR); should be Fourier transform infrared (FTIR) spectroscopy.

For the reader's better understanding, please make all the sub-figures into one figure in a bar format.

Provide the digital image of the agar plate of the bacterial suspension. Calculate its antibacterial activities in terms of %. Follow and cite this paper in the manuscript accordingly. Journal of the Taiwan Institute of Chemical Engineers 134 (2022) 104301.

All the equations are in a different format. Please rearrange it in the journal guidelines.

Section 3.4.1 discussion section needs to be improved. Insert the citations in the last sentence of this section. Calculate the relative crystallinity.

Present the delta H value in the DSC discussion section.

There are many unnecessary spaces.

Fig. 11 clarity is so low.

The mechanism of antibacterial needs to improve in a better way.

At present, the conclusion is too long. Please rewrite it preciously with important results.

The references section needs to be improved. There are no references from 2022. Please insert.

Round 2

Reviewer 1 Report

Many of the required amendments identified at the initial review have been made, and the paper is much improved as a result. There are still some outstanding issues though, which I set out below.

-          I think the authors discussion of the XRD data, and the % crystallinity calculated, is erroneous. The materials appear to be entirely amorphous by XRD, and my guess in the DSC is that there is some solvent loss at ca 60-65C and the peaks in the KP extract are evaporation or degradation of some sort.

-          Table 3/Supporting info – why have you fitted the zero-order model to data that are clearly non-linear? I would move Table 3 to supporting info in the interests of brevity.

-          Section 3.6 needs to go BEFORE the drug release data. You need the EE% to be able to calculate % release!

-          Lines 590-596 – the authors say that zero-order release is needed for the intended application – but their formulations do not give this. So surely this defeats the point of their study?

-          Table 5 should move to supporting info.

Reviewer 2 Report

Authors modified the manuscript as per my suggestions. Now it can be accepted for publication.
